# Ecological Livability Assessment of Urban Agglomerations in Guangdong-Hong Kong-Macao Greater Bay Area

**DOI:** 10.3390/ijerph182413349

**Published:** 2021-12-18

**Authors:** Zengzeng Fan, Yuanyang Wang, Yanchao Feng

**Affiliations:** 1Business School, Zhengzhou University, Zhengzhou 450001, China; fanzengzeng@zzu.edu.cn; 2Economics and Management School, Wuhan University, Wuhan 430072, China; 2017101050061@whu.edu.cn

**Keywords:** Guangdong-Hong Kong-Macao Greater Bay Area, ecological livable city, principal component analysis, spatial panel model

## Abstract

This paper proposes the “citizen-ecology-city” evaluation framework for urban ecological livability theoretically and studies the ecological livability of the Guangdong-Hong Kong-Macao Greater Bay Area (GBA) empirically. In addition, we analyze the factors of urban ecological livability in a spatial dynamic panel model. The results are as follows. (1) Ecological livability levels of Macao and Hong Kong are significantly higher than the nine cities in the PRD; (2) Shenzhen and Guangzhou lead the nine cities in the PRD, while Jiangmen and Zhaoqing perform poorly; (3) GBA cities can be divided into three categories: Macao, Hong Kong, Shenzhen, and Guangzhou in the first tier; Zhuhai, Foshan, and Dongguan in the second tier; Huizhou, Zhongshan, Jiangmen, and Zhaoqing in the third tier; and (4) The ecological livability of the GBA cities has a characteristic of spatial correlation. In terms of the international value, the three-dimensional evaluation framework can apply to other bay areas in the world.

## 1. Introduction

Continuous aggregation of socioeconomic factors is the critical issue of sustainable development in urban agglomerations [1]. The population concentration in cityscapes dominated by technology and built infrastructure has fostered an urban society that is increasingly decoupled and independent from ecosystems [2]. However, demands on natural capital and ecosystem services increase steadily in our urbanized planet [3,4,5]. On the other hand, urban ecosystems are an open frontier in ecosystem service research [6], ecosystem services are the benefits that human beings obtain directly or indirectly from the ecosystem, and ecological livability is a concept closely related to ecosystem services. In particular, environmental livability always connects with the city, and it focuses on ecological construction and planning, which means human beings are the behavioral subject. In addition, ecological construction and planning are not always reasonable, and people also consider non-ecological elements when evaluating a city’s environmental livability. 

Cities in the bay area where the economy is generally highly developed have experienced enormous economic growth and rapid urban transformation over the past few decades. The livability of the bay area has attracted the attention of some scholars [7]. The Guangdong-Hong Kong-Macao Greater Bay Area (GBA), the New York Bay Area, the San Francisco Bay Area, and the Tokyo Bay Area are known as the four Bay Areas in the world. The GBA, which consists of two special administrative regions (SAR), Hong Kong and Macao, and nine cities in the Pearl River Delta (PRD) (Figure 1), is a primary strategic concept put forward by the Chinese government which has both geographical and economic attributes [8]. GBA has a strong talent attraction and is one of China’s regions with the most significant population density for its high openness and solid economic vitality. On 56,000 square kilometers of land, the total population has reached 70 million at the end of 2018, and there is an incremental trend. Population agglomeration provides sufficient labor and reserves affluent intellectual resources for GBA and causes environmental pollution, energy shortage, traffic jams, and rising house prices. Li et al. [9] suggest prioritizing the sustainability objectives of major public projects in the GBA by evaluating various sustainability objectives from a multi-stakeholder perspective. Their findings indicate the need for more consideration of social concerns in Guangdong, in addition to the proper public participation in Hong Kong to avoid excessive interruptions to the pace of project procurement. And Macao may have to experience a relatively slow development of construction to balance the social/environmental requirements involved. To achieve green and sustainable development, people pay more attention to the ecological livability of urban development. This paper studies the ecological livability of the urban agglomeration in GBA to know about the construction of the ecological livable urban agglomeration in the GBA in order to provide some guidance and scientific basis for the construction of the world-class urban agglomeration in the GBA.

Hong Kong, a hybrid city, characterized by traditional compact urban neighborhoods in Kowloon and Hong Kong Island and transit-oriented new towns in the New Territories, comprises three major parts: Kowloon, Hong Kong Island, and the New Territories. Its urban developments that often achieved high density, mixed land-use, and gridiron street networks were built before WWII and concentrated in Hong Kong Island and Kowloon [10]. Hong Kong and Macao are famous for their developed economies in the world. However, compared with the Chinese mainland, which has experienced rapid development since the reform and opening-up, the speed of development in Hong Kong and Macao is relatively slower. One problem worrying people in Hong Kong and Macao is the high housing price and narrow living space. Livability is the core of building a high-quality living circle in the GBA, while the excessive housing costs burden has become an obstacle to livability [11]. The phenomenon motivates us to research whether Hong Kong and Macao are livable cities. In this paper, through establishing an ecological livability evaluation index system and comparing the livability score of GBA cities, we have found that Hong Kong and Macao are ecologically livable cities compared with geographically adjacent cities. According to the findings, economic development is a prerequisite for livability. A livable city always has a developed economy. Therefore, there is no distortion between urban ecological livability and economic development at the present stage. 

The marginal contributions of this paper are summarized as follows. Firstly, we propose the “citizen-ecology-city” three-dimensional theoretical framework for urban ecological livability and establish the evaluation index system. The theoretical framework and the evaluation index system apply to other urban agglomerations worldwide. Although there have been index systems in the existing literature, the existing index systems are either not comprehensive or not practical because of weak operability. Therefore, developing a new index system is necessary to evaluate urban ecological livability scientifically and comprehensively, especially for urban agglomerations in the bay area like the GBA, where two different institutions are implemented in a region, resulting in statistical differences in indicators. According to the index system established in this paper, we can compare the ecological livability of any two cities that belong to socialist and capitalist countries, respectively. Secondly, we evaluate the ecological livability of the urban agglomeration in the GBA quantitatively. For the difficulties in data collection and data processing of unified indicator caliber, we haven’t found any relevant literature on the ecological livability of urban agglomerations in the GBA. Thirdly, through a spatial econometric model, we found that the environmental livability of GBA urban agglomeration has a significant spatial correlation. It enlightens us that the urban agglomeration of GBA is a closely linked whole, which means that we should not only strengthen economic and trade cooperation and people-to-people exchanges but also cooperate in environmental protection and green sustainable development. Fourth, through empirical analysis, we divide the factors of urban ecological livability into three categories: key factors, essential factors, and general factors, which can guide the government to take targeted measures to improve the ecological livability of GBA urban agglomeration.

The rest of this paper is arranged as follows. The second part is the related literature review. The third part is the foundation of the evaluation index system of urban ecological livability. The fourth part is the empirical analysis of GBA urban agglomeration. Finally, we propose the research summary and policy recommendations.

## 2. Literature Review

Livable cities are of great significance today. One reason is that human beings need to live in healthier, greener, and more inclusive cities. However, some causes hinder cities from being livable. The poor-designed redevelopment of historic neighborhoods makes historic cities not comfortable [12]. Urban planning has been used as a “weapon” for capitalist interests [13]. A desirable urban environment should be resilient and more sustainable. Behind creating a better urban environment, among the results of urban planning as a generator and control mechanism, there are social and economic polarization, rising rents and housing costs, the displacement of people [14], high-standard new developments [15], and privatization. Unhealthy solutions for urban regeneration are seemingly trying to be overcome after the pandemic [16,17] to make cities more livable. People worldwide hope to live in comfortable urban areas and enjoy a high quality of life. At the same time, factors that are not conducive to livability can be seen everywhere in the international context. Therefore, we should know about the elements that make cities not livable and the factors that make cities comfortable. 

For the concepts of a livable city and urban ecological livability, there is no unified definition [18]. E. Salzano [19], D. Hahlweg [20], P. Evans [21], and other scholars have proposed their definitions, respectively. Comparing the meanings of these concepts (see Table 1), we find some commonalities in different senses. They all emphasize the long-term life and development needs of citizens. The body of the livable city is human, so humanity is at the core of a livable city, and its definition is also around the needs of human survival and development. Livability has contained the ecological requirements, and ecological livability emphasizes the importance of ecology in livable city planning and construction.

From the definition of a livable city, this paper defines the livable ecological city from three aspects: citizen, ecology, and city. Table 2 shows the detailed content. The citizen dimension is divided into the three elements of survival, development, and happiness. The ecological size is embodied in the environment, ecosystem, and development principle. The urban dimension includes two sub-items: comprehensive urban function and urban development level. The definition of a livable ecological city will provide the basis for developing the evaluation index system of urban ecological livability.

There have been researches on the evaluation of urban ecological livability. Lu et al. [27] constructed an evaluation index system of ecological livability from two aspects of natural ecology and human ecology and established a fuzzy matter-element model based on entropy theory to evaluate and analyze the ecological livability of the Central Plains city group. They found that the overall level of ecological livability of the Central Plains city group was not high, and the nine cities could be divided into three levels. Gong and Chen [28] developed an evaluation model of the livable ecological city from the four aspects of economic development, infrastructure construction, ecological environment, and social livelihood. They ranked the ecological livability of 16 cities in Anhui. The results found that the contribution of the air quality excellence rate and the proportion of the tertiary industry is relatively significant. Most studies take ecology as an aspect of livability, which is involved in the construction of the evaluation index system, and then evaluate the city’s livability, which lacks pertinence compared with the research of ecological livability. Cui et al. [29] established an evaluation index system from five aspects of comfort level, convenience level, happiness level, development level, and safety level, evaluated the livability level of urban agglomeration in Beijing, Tianjin, and Hebei, and analyzed its main factors. It is found that cities with the top three average livable levels are Beijing, Tianjin, and Qinhuangdao, excellent days of level 2 and above, the number of primary school students, and per capita urban road area have a significant positive impact on the livability. Quan and Liu [30] used the AHP decision-making analysis method to evaluate the livability of Chinese provincial capitals and municipalities from five aspects: the economic development level, ecological environment quality, social civilization level, living convenience level, and infrastructure construction. They found that the distribution of livable cities in China is unbalanced, and geographical location, economic development level, and education level have a significant impact on the livability of cities. Additionally, similar evidence is found in the example of rural China [31,32].

## 3. Evaluation Index System of Urban Ecological Livability

Hu and Hu [33] set up an index system for evaluating the livability of cities, including 81 indexes. However, there are many problems with the index system. First, the fourth level indexes do not provide specific data sources and are not operational, such as the transparency of government affairs and the integrity rate of lifeline projects. Second, some fourth-level indexes cannot be quantified and are ambiguous, such as urban characteristics, emergency response mechanisms for natural disasters, the regular supply of tap water, and efforts to crack down on fake and shoddy goods. Third, the indicators are miscellaneous and various. The same content is repeatedly measured, such as the per capita disposable income and per capita wage income, the number of college students per 10,000 people, and the number of graduate students. Also, the authors did not verify the feasibility and rationality of the index system with relevant data. Therefore, the index system of livability evaluation is more like the content system of livability. According to the content of the livable ecological city, referring to the domestic and international evaluation index system and the urban ecological livability evaluation index system constructed by Lu et al. [27], taking into account the principles of comprehensive, refined, and quantifiable, this paper establishes the urban ecological livable evaluation index system from the three dimensions of citizen, ecology, and city (Table 3). In the index system, the urban registered unemployment rate and the daytime average equivalent sound level of road traffic noise are negative indicators; the permanent population density is a neutral indicator (from the perspective of indication, the greater the population density, the higher the ecological livability of the city. From the standpoint of social convenience and social network development, the greater the population density, the higher the ecological livability of the city. But from the perspective of per capita social resources, living space, and urban congestion, the greater the population density, the worse the ecological livability. Therefore, we regard the resident population density as a neutral index.), and the rest are positive indicators.

## 4. An Empirical Analysis of Urban Agglomeration in GBA

### 4.1. Data

The data used in this paper are from statistical yearbooks of each city, bulletins of national economic and social development statistics, bulletins of environmental conditions, etc.; data of Hong Kong SAR are from the Hong Kong Statistical Yearbook, the website of the Statistics Office of Hong Kong SAR government, etc.; data of Macao SAR are from Macao statistical yearbooks, annual reports of the Macao Post and telecommunications, etc. For some cities, some indicators have missing values in some years, so the interpolation method is used to supplement. Finally, we obtain the balanced panel data of “9 + 2” cities in the GBA from 2010 to 2017. Descriptive statistics of urban panel data are presented along with the indicator system (see Table 3).

Since Hong Kong and Macao have a different monetary system from the Chinese mainland, we use the annual average exchange rate of that year to convert the Hong Kong dollar and the Macao dollar into RMB yuan to enhance comparability in a unified unit when measuring the urban development with per capita GDP and the per capita annual disposable income of urban residents. In addition, we also process other differences between cities caused by statistical calibration and statistical methods in the data collection and collation.

### 4.2. Evaluation Method

The first step of evaluating the ecological livability of cities in the GBA is to get a comprehensive index of urban ecological livability. In previous research, the popular methods of analysis used are the AHP decision analysis method [30], principal component analysis (PCA) [34,35,36], fuzzy matter-element model evaluation analysis [26], etc. Among them, PCA is widely used [37] because it can evaluate the objective economic phenomena scientifically by calculating the score of the principal component function. Therefore, this paper uses the PCA method to assess the ecological livability of cities in the GBA.

To highlight the specification of PCA, we introduce some formal notation and nomenclature. *X* is a matrix of data with *I* rows and *J* columns. x*_j_* is the individual variable of *X* in the form of a vector with *I* dimensions. *t* = *w*_1_*x*_1_ + … + *w_J_x_J_* = *Xw*. ${\left\|\cdot \right\|}^{2}$ indicates the squared Frobenius norm. The problem is
(1)$$\underset{\|w\|=1}{\mathrm{argmax}}\;var\left(t\right)=\underset{\|w\|=1}{\mathrm{argmax}}\;\left(t^{T}t\right)=\underset{\|w\|=1}{\mathrm{argmax}}\;\left(w^{T}X^{T}Xw\right)$$

To calculate how representative *t* is in replacing *X*, we regress all *X* variables on *t* using the ordinary regression equation.
(2)$$X=tp^{T}+E=\hat{X}+E$$
where *p* is the vector of coefficient and E is the matrix of residuals. The quality of *t* is judged by calculating $\frac{{\left\|X\right\|}^{2}-{\left\|E\right\|}^{2}}{{\left\|X\right\|}^{2}}\times 100\%$.

From Equation (2), both *t* and *p* can be established by solving $\underset{t,p}{\mathrm{argmin}}\|X-tp^{T}{\|}^{2}$. The vector *t* is usually referred to as the score vector and the vector *p* is called the loading vector.

We use their reciprocal to make them favorable for the negative index of the urban registered unemployment rate and road traffic noise daytime average equivalent sound level. Considering the different units of indicators and the different magnitude of sample values of variables, we standardize the data first and then carry out the analysis to eliminate their impact on the PCA. For the standardized data, the mean of each variable is 0, and the variance is 1.

When analyzing the factors of urban ecological livability, we establish the following general dynamic spatial panel model [38], and then the estimation is carried out:(3)$$\{\begin{array}{c} y_{it}=\tau y_{i,t-1}+\rho w_{i}{}^{\prime }y_{t}+x_{it}{}^{\prime }\beta +d_{i}{}^{\prime }X_{t}\delta +u_{i}+\gamma_{t}+\varepsilon_{it}\\ \varepsilon_{it}=\lambda m_{i}{}^{\prime }\varepsilon_{t}+v_{it} \end{array}$$

The explained variable $y_{it}$ is the ecological livability level of the city *i* in year t, which is expressed by the comprehensive score of ecological livability; $y_{i,\;t-1}$ is the first-order lag of the explained variable; $\rho w_{i}{}^{\prime }y_{t}$ is the spatial lag term of the explained variable; $x_{it}{}^{\prime }$ is the influencing factor vector of ecological livability; $d_{i}{}^{\prime }X_{t}\delta$ is the spatial lag of the explanatory variable; $u_{i}$ is the individual effect of city *i*, and $\gamma_{t}$ is the time effect, $m_{i}{}^{\prime }$ is the line *i* of the error term space weight matrix. The spatial weight matrix is a symmetric matrix *W* which is composed of 0–1 elements based on the geographical adjacent relationships.

### 4.3. Evaluation Results

We analyze the standardized data with PCA. As a result, nine components with eigenvalues greater than 1 (9.81, 5.23, 2.70, 2.35, 2.05, 1.71, 1.59, 1.27, 1.18) contain 84.54% information of the original variables and could replace the actual variables well. Therefore, nine principal components are selected to evaluate the ecological livability of cities. Finally, we obtain the total score and ranking of ecological livability of GBA urban agglomeration (Table 4).

Macao, Hong Kong, Shenzhen, Guangzhou, and Zhuhai are in the top five positions, and Zhaoqing has always been the bottom of the GBA urban agglomeration. According to the scores, GBA cities can be roughly divided into three categories: Macao, Hong Kong, Shenzhen, and Guangzhou; Zhuhai, Foshan, and Dongguan; Huizhou, Zhongshan, Jiangmen, and Zhaoqing. Macao, Hong Kong, Shenzhen, and Guangzhou belong to the first-tier. Their total scores are always greater than 0 during the period, much higher than that of other cities. The second-tier cities, including Zhuhai, Foshan, and Dongguan, have significantly increased their total scores, turning negative into positive from 2012, 2016, and 2015. In the third-tier cities, except for Zhongshan, whose total score was slightly greater than 0 in 2017, their scores are always negative. Huizhou’s total score is rising weakly, and Zhaoqing is far from the turning point of negative into positive.

Figure 2 shows the time trend of total scores of ecological livability of cities in the GBA. The ecological livability of the GBA cities is rising during the studied period. The in-depth analysis shows that Foshan and Dongguan have the fastest rising speed, close to a straight line. Hong Kong and Guangzhou are stable and positive. Macao and Zhuhai tend to be stable after rising to a certain level, while Shenzhen and Huizhou grow with slight fluctuations. Jiangmen and Zhaoqing crawl up with ample space for improvement. After a downturn, Zhongshan has accelerated and entered a period of rapid development. In summation, the overall ecological livability of urban agglomeration in the GBA is relatively high and shows a steady upward trend.

### 4.4. Evaluation of the Factors of Urban Ecological Livability

In the evaluation index system established above, each three-level index will affect some aspects of urban ecological livability to a certain extent. However, their impact on environmental livability is heterogeneous. Some indexes have a more significant effect on urban ecological livability than others. When the government undertakes measures to improve environmental livability, it is impossible to do everything under the limited financial, material, and human resources. Considering all aspects affecting ecological livability, we can only make breakthroughs in critical areas. The following empirical analysis is carried out through an econometric model to identify the key factors, essential factors, and general factors from the evaluation index system’s elements and understand the critical point of improving ecological livability.

According to the ranking of ecological livability of GBA cities, we find the following characteristics: the geographical locations of the top cities are relatively close, such as Hong Kong and Shenzhen. Therefore, we should first investigate whether there is spatial dependence in urban data to decide whether to adopt the spatial method. The spatial autocorrelation test (Table 5) on the total scores of ecological livability of cities in the GBA shows a spatial autocorrelation indeed, so we should adopt the spatial measurement method. And according to Moran‘s I (positive) and Geary‘s C (less than 1) [39], there is a positive autocorrelation.

Based on the tests, we should use the spatial panel model. The ecological livability level of a city is the explained variable. There is usually a temporal lag effect. The ecological livability level in the current year will have a specific dependence on the environmental livability level in the last year. Thus we should employ the spatial dynamic panel model. If the coefficient of the first-order lagged explained variable is close to 0 or not significant, there is no lag effect, and the setting of the dynamic panel is still adequate. Besides, although ecological livability has spatial effects, it is unclear how this spatial dependence occurs. For example, we do not know whether the ecological livability level of the city *i* is affected by the explanatory variables of city *j* or whether there is a spatial correlation in the possible missing variables or unobservable random shocks. Therefore, to avoid the estimation error caused by the unreasonable pre-set model, we furthermore adopt the general dynamic spatial panel model in Equation (3) [38].

Special cases of model (3), the spatial Durbin model (SDM), the spatial autocorrelation (SAC) model, and the spatial error model (SEM) are estimated [40]. All the explanatory variables used in the estimation are standardized data. Therefore, there is no clear economic meaning for the size of the estimated coefficient. Consequently, we only pay attention to the significance of explanatory variable coefficients and classify explanatory variables according to their significance level. Table 6 shows the results. When estimating SDM, the Hausman test shows that the null hypothesis of random effect is acceptable, so the individual random effect model in column (1) is calculated. We only use the fixed effect when calculating the SAC model, so the individual fixed effect model in column (2) is evaluated. Columns (3) and (4) are SEM random effect and time fixed effect estimation results, respectively.

According to the estimation results of the three spatial panel models, the spatial lag coefficients *ρ* (rho) and *λ* (lambda) have good significance, indicating that there is indeed a spatial effect, and the model setting is correct. Comparing the significance level of the spatial coefficients in columns (1) and (2), we can see that the spatial lag effect of the interpreted variable *y* is mainly caused by the spatial correlation of the omitted variables. The omitted variables impact *y* or the spatial correlation of the unobservable random shocks. $m_{i}{}^{\prime }\varepsilon_{t}$ plays a full intermediary role between $w_{i}{}^{\prime }y_{t}$ and $y_{it}$. Most explanatory variables of the four estimations have the same significance level, and the estimated values are very close, which has good robustness.

### 4.5. Robustness Analysis

In the spatial panel model, the popular spatial weight matrix is the symmetric matrix established based on the adjacent relationships without considering the heterogeneity of the neighboring associations. The adjacent relationships are the same, but some of the adjacent relationships are closely related, some of them are relatively distant, and some of them are still affected by the radiation of non-adjacent cities. Next, we use another spatial weight matrix to analyze the robustness of the estimation results above, and the following spatial weight matrix is constructed based on the traffic accessibility,
(4)$$W_{1}=\left\{w_{ij}=\frac{1}{t_{ij}}\right\}$$
where $t_{ij}$ is the least running time of railway, highway, and other transportation means between city *i* and city *j.* It mainly refers to the operation time of high-speed rail between two cities. We take the minimum of other train and bus operation times for two cities without high-speed rail. The spatial weight matrix considers the heterogeneity of the connection between cities and uses the matrix to re-estimate the spatial panel model above. Table 7 shows the results. For SDM and SEM, the Hausman test shows that the null hypothesis of random effect can still be accepted, and the replacement of the spatial weight matrix does not affect the test results. As a comparison, column (4) gives the estimation results of the SEM with individual fixed effect, which is very close to the random effect estimation results in column (3). Comparing the spatial coefficients in columns (1) and (2), $m_{i}{}^{\prime }\varepsilon_{t}$ has changed from the full mediating effect (Table 6) to the partial mediating effect between $w_{i}{}^{\prime }y_{t}$ and $y_{it}$. Comparing Table 6 with Table 7, except for the significant change of spatial lag coefficient of explanatory variables in SDM in column (1), the significance of most explanatory variables is almost unchanged, and the estimated values are very close. Therefore, the estimation results have good robustness. The significant change of spatial lag coefficient of the explanatory variable shows that it is sensitive to the selection of the spatial weight matrix.

According to the estimation of the above models, we divide the factors of urban ecological livability into three categories: key factors, essential factors, and general factors. The key factors refer to the factors that significantly impact urban ecological livability and have a significant spatial lag effect. The essential factors refer to the factors that have a very substantial impact on urban ecological livability. Finally, the general factors refer to the factors whose effect on the level of urban ecological livability is not very significant or not significant. Based on the classification, there are 18 key factors: social endowment insurance coverage rate, urban registered unemployment rate, proportion of education expenditure to financial expenditure, number of cinemas, bus ownership per 10,000 people, daytime average equivalent sound level of road traffic noise, sewage treatment rate, comprehensive utilization rate of industrial solid waste, harmless treatment rate of domestic waste, per capita urban road area, water use penetration, density of drainage network, proportion of output value of tertiary industry in GDP, GDP growth rate, per capita GDP, per capita disposable income of urban residents, comprehensive population coverage rate of radio, and comprehensive population coverage rate of television; 13 essential factors: per capita living area of urban residents, the density of permanent population, the number of students in Colleges and universities per 10,000 people, the ratio of teachers and students in primary schools, the density of postal outlets, and telephone number (including mobile phone) penetration rate, the number of Internet users, the air quality compliance rate, per capita water resources, per capita public library collection, green coverage rate of built-up area, per capita park green space area, per 10,000 invention patents authorized; and general factors include the proportion of real estate investment in GDP and the number of hospital beds per 10,000 people. Therefore, most of the factors of urban ecological livability are key and essential factors, indicating that there are almost no irrelevant variables, which further verifies the effectiveness of the evaluation index system of urban ecological livability.

## 5. Conclusions

### 5.1. Findings

Based on the above research work, the main findings are as follows. (1) In the GBA, the ecological livability of Hong Kong and Macao is much higher than that of the nine cities in the PRD, and first-tier cities Shenzhen and Guangzhou are in the leading position of the nine cities in the PRD; (2) In the three dimensions that affect the ecological livability of the GBA cities, from the perspective of citizens, people pay more attention to personal development opportunities and happiness; from the ecology aspect, the living environment and ecosystem have a significant impact on the ecological livability; from the urban dimension, the comprehensive functions and development level of the city have a substantial effect on the ecological livability; and (3) Although Hong Kong and Macao are crowded and people’s living space is minimal, they are ecologically livable cities in terms of the comprehensive perspective of civil development opportunities, ecological environment and ecosystem, comprehensive urban functions, and economic and social development.

### 5.2. Policy Implications

We put forward the following policy recommendations according to the research findings above: (1) Based on the key factors affecting the livable ecological level of the city, we should make efforts to improve the comfortable ecological level of Jiangmen, Zhaoqing, and other cities from three aspects of citizen, ecology, and city to improve the overall ecological livable level of GBA urban agglomeration. The results of the ecological livability evaluation of the GBA urban agglomeration show that the ecological livability levels of Jiangmen and Zhaoqing are at the bottom of the GBA urban agglomeration, far below the average level, which seriously restricts the overall improvement of ecological livability of the GBA. Therefore, we should take targeted measures to improve their environmental livability; (2) Economic development comes first for the nine cities in the PRD. After all, economic growth can bring more livability [41]. Therefore, take economic construction as the center, and urban construction should learn more advanced experience from Hong Kong and Macao. Macao and Hong Kong are the most crowded cities in the GBA, but this does not make them unlivable. The reason is that Macao and Hong Kong are the top two cities in GBA in terms of per capita GDP. However, even from a global perspective, Macao and Hong Kong ranked fourth and 22nd in the global economy in 2018. As for per capita PPP GDP, Macao, Hong Kong ranked second and 11th in the global economy in 2018, respectively (World Bank, Washington, DC, USA).

The high level of ecological livability is inseparable from the developed economy. Considering infrastructure, Macao and Hong Kong have a narrow geographical area for urban construction. The width of urban roads is generally thinner than that of nine cities in the PRD. Still, the frequency of traffic jams is not high, which shows that the operation efficiency of traffic infrastructure in Hong Kong and Macao is very high. Regarding drainage systems and sewage treatment systems, the GBA is located in the south of China, where there is a lot of rain. Still, Hong Kong and Macao rarely suffer from flood disasters, and there is a relatively faultless sewage treatment and recycling system. In these aspects, nine cities in the PRD should learn from their successful experiences. (3) While deepening the economic cooperation between Guangdong, Hong Kong, and Macao, we should strengthen the cooperation in ecological protection and environmental governance and jointly build a comfortable and livable world-class city group with green development and sound ecology. This paper finds a spatial correlation between ecological livability and its key factors. Therefore, strengthening the cooperation of ecological environment protection among cities can promote each other and form a joint force to realize the sustainable development of GBA better and faster.

### 5.3. Discussion

Though the evaluation index system established in this paper has good application to the GBA through empirical study, it has not been used to analyze the ecological livability of other bay areas because of the lack of data. In future research, a domestic or international comparative study of urban environmental livability can be conducted based on the evaluation index system of urban ecological livability. For example, we can compare GBA urban agglomeration with Yangtze River Delta urban agglomeration and Beijing-Tianjin-Hebei urban agglomeration for domestic comparison. In addition, we can compare the GBA with the Tokyo Bay Area, New York Bay Area, and San Francisco Bay Area for international comparison. More empirical research will test the adequacy of the evaluation index system and verify the theoretical framework of urban ecological livability proposed in this paper.

## Figures and Tables

**Figure 1 ijerph-18-13349-f001:**
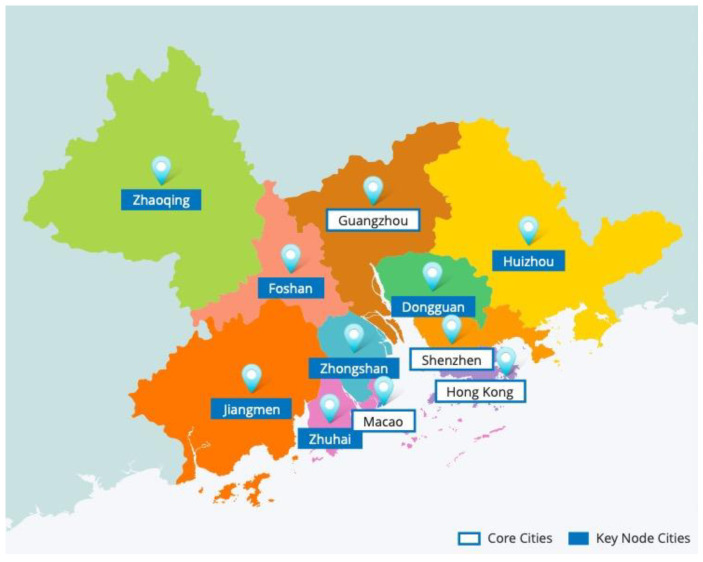
GBA Cities.

**Figure 2 ijerph-18-13349-f002:**
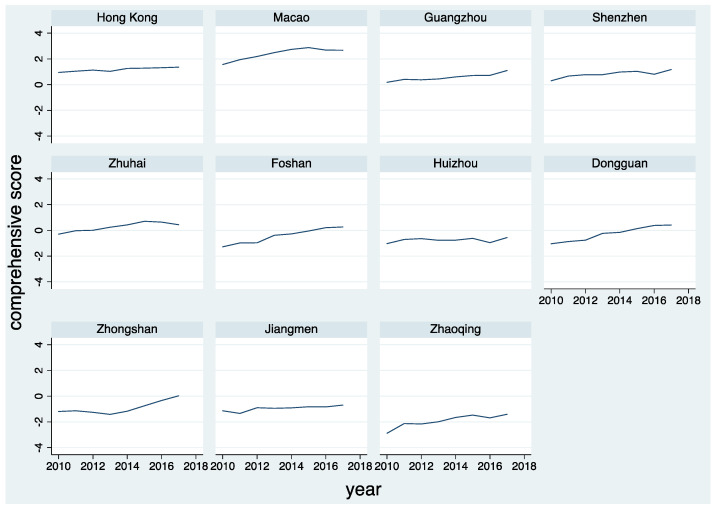
Time trend of ecological livability comprehensive score of urban agglomeration in GBA. Source: Plotted by the authors.

**Table 1 ijerph-18-13349-t001:** Summary of definition of a livable city and its related concepts.

Concept	Source of Definition	Defining Points
Livable city	the long-term plan for Greater Vancouver (2003)	To bring physical, psychological and social benefits, personal development opportunities and rich spiritual and cultural wealth to the public; important principles: fairness, dignity, accessibility, cheerfulness, participation and rights protection [22].
Urban Livability	P. Evans (2002)	Survival and ecological sustainability; meet the survival needs of all citizens on the premise of protecting the ecological environment.
Livable city	D. Hahlweg (1997)	Healthy life, convenient travel, safe and charming, shared by all.
Livable city	E. Salzano (1997)	Connecting history and future, respecting historical footprints and future generations, sustainable development, providing material and social welfare, public space is the center of community and social life, and a network extending from downtown to suburb.
Livable city movement	Timothy D. Berg (1999)	Reshape the urban environment, build roads and blocks suitable for pedestrians, realize the comprehensive functions of the city such as work, residence and retail, and enhance the diversity of the city [23].
Livable City	H.L. Lennard (1997)	Citizens feel the existence of each other and face-to-face communication. Citizens are involved in many activities and celebrations. They feel safe. Public space can be used as a learning place and each citizen can be a learning object. It has multiple functions such as economy, society and culture. Citizens respect each other, the urban environment has an aesthetic feeling, the opinions of citizens are respected and can participate in the process of urban development [24].
Evaluation index system of urban ecological livability	Lu et al. (2012)	It covers the connotation of urban sustainable development strategic objectives, comprehensively reflects the stability and health of the urban ecosystem, as well as all aspects of economic development, social development and ecological environment quality, objectively and truly reflects the urban ecological livability, and conforms to the concept, objectives and evaluation principles of urban sustainable development.
Livable City	Zhang (2016)	A livable city should be a city with a healthy environment, safety, pleasant nature, harmonious society, convenient life, and convenient travel [25].
Livable City	Liu et al. (2019)	Livable City is a city with high quality of life and comfortable and pleasant living [26].

Source: The authors sorted out.

**Table 2 ijerph-18-13349-t002:** Connotation of an ecological livable city.

Ecological livable city	citizen	Survival	All survival needs such as security are met, and medical and pension systems are improved.
		development	Fair, respected, power guaranteed, participation in urban development process, personal development opportunities
		happiness	citizens can obtain rich spiritual and cultural wealth, comfortable mood, harmonious neighborhood and convenient life.
	ecology	environment	Beautiful living environment, healthy and clean air, water and soil
		ecosystem	Stable and healthy
		development principles	Green, environmental protection and sustainability
	city	comprehensive functions	It has multiple functions such as work, residence and retail, and its infrastructure construction is perfect.
		Development level	developed economy, harmonious society and prosperous culture

Source: Summarized by the authors.

**Table 3 ijerph-18-13349-t003:** Evaluation index system of urban ecological livability.

Comprehensive Index	Dimension	First Level Index	Second Level Index	Proxy Variable	Mean	S.D.
urban ecological livability	Citizen	Survival	Living space	Per capita living area of urban residents (m^2^)	30.93	12.95
			Accessibility	Permanent resident population density (person/km^2^)	3926.48	5493.42
			Residential quality	Proportion of real estate investment in GDP (%)	10.98	6.23
			Medical conditions	Number of beds in hospital per 10,000 people	40.25	15.87
			Pension services	Social endowment insurance coverage rate (%)	51.79	24.21
		Development	Employment opportunities	Registered urban unemployment rate (%)	2.38	0.39
			Education expenditure	Ratio of education expenditure to total financial expenditure (%)	18.60	5.03
			Talent density	Number of college students per 10,000	307.22	345.81
			Faculty	Ratio of teachers to students in primary school	0.05	0.01
		Happiness	Spiritual culture	Number of cinemas	38.88	31.80
			Life convenience	Density of postal outlets (PCs./km^2^)	0.17	0.18
				Urban telephone penetration rate (including mobile phone) (Department/100 people)	341.58	166.47
				Number of Internet users (10,000)	255.54	186.06
				Public transport vehicles per 10,000 people	11.39	5.64
	Ecology	Environment	Natural environment	Air quality compliance rate (%)	90.25	8.96
				Water resources per capita (m^3^)	591.05	909.10
				Average equivalent sound level of road traffic noise in daytime (DB)	68.06	1.07
			Cultural environment	Per capita public library collection (volume)	1.31	1.01
			Environmental improvement	Sewage treatment rate (%)	92.58	6.44
				Green coverage rate of built-up area (%)	46.10	10.72
				Per capita park green area (m^2^)	11.92	6.47
		Ecosystem	Health	Comprehensive utilization rate of industrial solid waste (%)	88.53	15.77
				Harmless treatment rate of urban domestic waste (%)	93.57	13.92
	City	Comprehensive function	Innovation	Number of invention patents authorized per 10,000 people	3.96	4.05
			Infrastructure	Urban Road area per capita (m^2^)	10.12	7.50
				Water use penetration (%)	98.72	2.65
				Density of drainage network (km/km^2^)	6.75	4.98
		Development level	Economic	Proportion of output value of tertiary industry in GDP (%)	55.23	20.13
				GDP growth rate (%)	9.33	5.05
				Logarithm GDP per capita (yuan)	11.52	0.73
				Logarithm Per capita disposable income of urban residents (yuan)	10.83	0.93
			Culture	Broadcast comprehensive population coverage (%)	99.98	0.08
				TV comprehensive population coverage (%)	99.96	0.17

Source: Calculated by the authors.

**Table 4 ijerph-18-13349-t004:** Comprehensive score and ranking of ecological livability of GBA Urban Agglomeration in 2010–2017.

City	Year							
	2010		2011		2012		2013	
	Score	Ranking	Score	Ranking	Score	Ranking	Score	Ranking
Hong Kong	0.9445	2	1.0455	2	1.1284	2	1.0398	2
Macao	1.5667	1	1.9394	1	2.1950	1	2.4926	1
Guangzhou	0.1791	4	0.4091	4	0.3811	4	0.4461	4
Shenzhen	0.2966	3	0.6677	3	0.7743	3	0.7714	3
Zhuhai	−0.2980	5	−0.0224	5	0.0046	5	0.2484	5
Foshan	−1.2778	10	−0.9757	8	−0.9597	9	−0.3805	7
Huizhou	−1.0298	6	−0.7071	6	−0.6351	6	−0.7691	8
Dongguan	−1.0432	7	−0.8670	7	−0.7637	7	−0.2304	6
Zhongshan	−1.1868	9	−1.1407	9	−1.2591	10	−1.4238	10
Jiangmen	−1.1398	8	−1.3420	10	−0.8967	8	−0.9427	9
Zhaoqing	−2.8853	11	−2.1358	11	−2.1687	11	−1.9938	11
continue	2014		2015		2016		2017	
Hong Kong	1.2676	2	1.2857	2	1.3208	2	1.3573	2
Macao	2.7424	1	2.8787	1	2.6834	1	2.6752	1
Guangzhou	0.6082	4	0.7071	4	0.7274	4	1.1051	4
Shenzhen	0.9852	3	1.0320	3	0.8080	3	1.1779	3
Zhuhai	0.4309	5	0.7070	5	0.6441	5	0.4435	5
Foshan	−0.2774	7	−0.0410	7	0.2092	7	0.2696	7
Huizhou	−0.7595	8	−0.6168	8	−0.9526	10	−0.5573	9
Dongguan	−0.1520	6	0.1377	6	0.3852	6	0.4139	6
Zhongshan	−1.1662	10	−0.7519	9	−0.3340	8	0.0284	8
Jiangmen	−0.9113	9	−0.8244	10	−0.8330	9	−0.6904	10
Zhaoqing	−1.6603	11	−1.4700	11	−1.6841	11	−1.4046	11

Source: Calculated by the authors.

**Table 5 ijerph-18-13349-t005:** Spatial autocorrelation test.

Spatial Autocorrelation Index		Z	*p*-Value	Result
Moran’s I	0.074	3.251	0.001	spatial autocorrelation exists
Geary’s C	0.547	−6.871	0.000	spatial autocorrelation exists
Getis-Ord G	−1.615	−3.251	0.001	spatial autocorrelation exists

Source: Calculated by the authors.

**Table 6 ijerph-18-13349-t006:** Spatial panel model estimation.

Explanatory Variable (x)	(1)	(2)	(3)	(4)
	Sdm_Re (Wx)	Sac_Fe	Sem_Re	Sem_Fe_Time
Residential area per capita	−0.04 *** (−0.01 **)	−0.09 ***	−0.06 ***	−0.05 ***
Resident population density	0.08 *** (−0.05 **)	−0.04	0.12 ***	0.12 ***
Proportion of real estate investment	0.00 (−0.01)	0.02 **	0.00	0.00
Hospital beds per 10,000 people	0.02 * (−0.01)	0.02	0.01	0.02 ***
Participation rate of social endowment insurance	0.02 *** (−0.00)	0.03 ***	0.04 ***	0.04 ***
registered urban unemployment rate	−0.04 *** (−0.01 ***)	−0.05 ***	−0.06 ***	−0.06 ***
Proportion of education expenditure	−0.11 *** (−0.03 ***)	−0.12 ***	−0.11 ***	−0.10 ***
Number of college students per 10,000	0.05 *** (0.05 ***)	0.05	0.06 ***	0.05 ***
Ratio of teachers to students in primary school	0.07 *** (0.03 ***)	0.08 ***	0.04 ***	0.06 ***
Number of cinemas	0.04 *** (0.01 ***)	0.04 ***	0.04 ***	0.04 ***
Density of postal outlets	0.09 *** (0.03 ***)	0.13 ***	0.11 ***	0.11 ***
Telephone penetration	0.08 *** (0.03 ***)	0.07 ***	0.06 ***	0.06 ***
Number of Internet users	−0.04 *** (−0.02 ***)	−0.06 ***	−0.03 ***	−0.03 ***
Buses for every 10,000 people	0.12 *** (0.02 ***)	0.12 ***	0.11 ***	0.11 ***
Air quality compliance rate	0.01 *** (−0.00 **)	0.00 ***	0.00 **	0.01 ***
Water resources per capita	−0.02 *** (0.01 ***)	−0.02 ***	−0.02 ***	−0.02 ***
Traffic noise	0.05 *** (0.02 ***)	0.05 ***	0.05 ***	0.04 ***
Per capita public library collection	0.09 *** (0.01 **)	0.09 ***	0.12 ***	0.14 ***
Sewage treatment rate	0.12 *** (0.03 ***)	0.11 ***	0.11 ***	0.12 ***
Green coverage of built-up area	0.07 *** (0.02 *)	0.11 ***	0.07 ***	0.07 ***
Per capita park green area	−0.03 *** (−0.02 ***)	−0.03 ***	−0.05 ***	−0.03 ***
Utilization rate of industrial solid waste	0.08 *** (0.02 ***)	0.09 ***	0.08 ***	0.08 ***
Harmless treatment rate of domestic waste	0.06 *** (0.01 ***)	0.07 ***	0.06 ***	0.07 ***
Patent authorization per 10,000 people	0.06 *** (0.01 ***)	0.06 ***	0.08 ***	0.08 ***
Per capita urban road area	0.05 *** (0.02 ***)	0.05 ***	0.05 ***	0.05 ***
Water use penetration	0.09 *** (0.02 ***)	0.08 ***	0.08 ***	0.09 ***
Density of drainage network	0.08 *** (0.03 ***)	0.08 ***	0.08 ***	0.09 ***
Proportion of output value of tertiary industry	0.13 *** (0.08 ***)	0.09 ***	0.09 ***	0.07 ***
GDP growth rate	−0.09 *** (−0.03 ***)	−0.09 ***	−0.08 ***	−0.08 ***
Per capita GDP	0.13 *** (0.03 ***)	0.12 ***	0.13 ***	0.13 ***
Per capita disposable income of citizens	0.12 *** (0.07 ***)	0.14 ***	0.09 ***	0.12 ***
Broadcast coverage	0.08 *** (0.02 ***)	0.08 ***	0.08 ***	0.08 ***
TV coverage	0.07 *** (0.01 ***)	0.06 ***	0.06 ***	0.06 ***
Spatial coefficient				
rho	−0.25 ***	−0.00		
lambda		−0.40 ***	−0.11 *	−0.19 ***

Note: *, **, and *** indicate that the significance levels of the coefficient are 10%, 5%, or 1%, respectively. Column (1) in brackets is the coefficient of the lag term of the explanatory variable, that is, the vector δ in model (1). Source: Calculated by the authors.

**Table 7 ijerph-18-13349-t007:** Robust analysis of spatial panel model estimation.

Explanatory Variable (x)	(1)	(2)	(3)	(4)
	Sdm_Re (Wx)	Sac_Fe	Sem_Re	Sem_Fe_Time
Residential area per capita	−0.04 *** (−0.23)	−0.05 ***	−0.07 ***	−0.06 ***
Resident population density	0.10 *** (0.51)	0.07 ***	0.11 ***	0.08 **
Proportion of real estate investment	−0.01 *** (−0.14)	−0.00	−0.01	−0.01
Hospital beds per 10,000 people	0.01 (−0.14 *)	0.03 **	0.01	0.00
Participation rate of social endowment insurance	0.05 *** (0.41 ***)	0.04 ***	0.03 ***	0.03 ***
Registered urban unemployment rate	−0.06 *** (−0.15 ***)	−0.06 ***	−0.06 ***	−0.06 ***
Proportion of education expenditure	−0.11 *** (−0.69 ***)	−0.12 ***	−0.11 ***	−0.11 ***
Number of college students per 10,000	0.06 *** (0.39 *)	0.10 ***	0.06 ***	0.09 ***
Ratio of teachers to students in primary school	0.03 ** (−0.12)	0.06 ***	0.05 ***	0.05 ***
Number of cinemas	0.04 *** (0.19 ***)	0.04 ***	0.04 ***	0.04 ***
Density of postal outlets	0.07 *** (−0.04)	0.10 ***	0.12 ***	0.12 ***
Telephone penetration	0.05 *** (0.31 *)	0.04 ***	0.06 ***	0.05 ***
Number of Internet users	−0.03 *** (−0.18 **)	−0.04 ***	−0.04 ***	−0.04 ***
Buses for every 10,000 people	0.11 *** (0.62 ***)	0.11 ***	0.11 ***	0.11 ***
Air quality compliance rate	0.00 *** (0.04 **)	0.00 *	0.00	0.00 **
Water resources per capita	−0.02 *** (0.08 *)	−0.02 ***	−0.02 ***	−0.02 ***
Traffic noise	0.05 *** (0.30 ***)	0.04 ***	0.05 ***	0.04 ***
Per capita public library collection	0.11 *** (0.11)	0.13 ***	0.12 ***	0.12 ***
Sewage treatment rate	0.12 *** (0.48 ***)	0.11 ***	0.11 ***	0.11 ***
Green coverage of built-up area	0.07 *** (0.38 **)	0.08 ***	0.07 ***	0.08 ***
Per capita park green area	−0.05 *** (−0.02)	−0.04 ***	−0.04 ***	−0.04 ***
Utilization rate of industrial solid waste	0.08 *** (0.38 ***)	0.08 ***	0.08 ***	0.08 ***
Harmless treatment rate of domestic waste	0.06 *** (0.27 ***)	0.06 ***	0.06 ***	0.07 ***
Patent authorization per 10,000 people	0.08 *** (0.04)	0.08 ***	0.08 ***	0.08 ***
Per capita urban road area	0.06 *** (0.23 ***)	0.06 ***	0.05 ***	0.05 ***
Water use penetration	0.08 *** (0.40 ***)	0.09 ***	0.09 ***	0.09 ***
Density of drainage network	0.07 *** (0.45 ***)	0.08 ***	0.08 ***	0.08 ***
Proportion of output value of tertiary industry	0.11 *** (0.69 ***)	0.08 ***	0.09 ***	0.08 ***
GDP growth rate	−0.09 *** (−0.42 ***)	−0.08 ***	−0.08 ***	−0.08 ***
Per capita GDP	0.12 *** (0.55 ***)	0.13 ***	0.13 ***	0.12 ***
Per capita disposable income of citizens	0.17 *** (1.76 ***)	0.13 ***	0.10 ***	0.13 ***
Broadcast coverage	0.08 *** (0.29 ***)	0.08 ***	0.08 ***	0.08 ***
TV coverage	0.07 ***(0.36 ***)	0.06 ***	0.06 ***	0.06 ***
Spatial coefficient				
rho	−4.92 ***	−0.34 ***		
lambda		−7.28 ***	−5.23 **	−7.17 ***

Note: *, **, and *** indicate that the significance levels of the coefficient are 10%, 5% or 1% respectively. Source: Calculated by the authors.

## Data Availability

The data used to support the findings of this study are available from the corresponding author upon request.

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
