# Peer review of "Ecological Livability Assessment of Urban Agglomerations in Guangdong-Hong Kong-Macao Greater Bay Area"

_ijerph, 2021, doi:10.3390/ijerph182413349_

Round 1

Reviewer 1 Report

“Ecological Livability” is not a good choice of keywords, as it does not relate well to the current literature discussion. Some better choices to relate to the emerging topics include nature-based solutions, ecosystem services, and quality-of-life:

https://www.sciencedirect.com/science/article/pii/S2213305421000205

https://www.sciencedirect.com/science/article/abs/pii/S0378778819337922

https://www.sciencedirect.com/science/article/pii/S016920462100178X

https://www.sciencedirect.com/science/article/abs/pii/S1470160X19301682

https://www.sciencedirect.com/science/article/abs/pii/S092180091200362X

This paper too many narrowly-focused case descriptions, and reads like better fit for a Chinese journal. For an international journal, the paper should have clearer theoretical and international contributions.

Abstract: There too many narrowly-focused case descriptions. For an international journal, the paper should have clearer theoretical and international contributions.

Line 27: “Guangdong-Hong Kong-Macao Greater Bay Area (GBA)” - start with international discussion rather than local cases, or international readers will lose interests immediately. There are abundant international discussions on livability and ecosystem services.

Line 110: There is no literature within about 10 years, which highlight the need to modify the papers keyword choice.

Line 200: “Therefore, this paper uses the principal component analysis method to evaluate the ecological livability of cities in the GBA”—the description is too rough. Please highlight the specifications of PCA.

Equation 1: Equations should be illustrated in the method section, not in the result section.

Line 366: “In this paper, three aspects of work are done.”—do not reiterate your method again. Be straightforward into the findings.  

Author Response

Responses to Reviewer # 1

Comments and Suggestions for Authors

“Ecological Livability” is not a good choice of keywords, as it does not relate well to the current literature discussion. Some better choices to relate to the emerging topics include nature-based solutions, ecosystem services, and quality-of-life:

https://www.sciencedirect.com/science/article/pii/S2213305421000205

https://www.sciencedirect.com/science/article/abs/pii/S0378778819337922

https://www.sciencedirect.com/science/article/pii/S016920462100178X

https://www.sciencedirect.com/science/article/abs/pii/S1470160X19301682

https://www.sciencedirect.com/science/article/abs/pii/S092180091200362X

Reply: Many thanks for your constructive suggestion. Although “ecological livability” is important in this study, it indeed does not relate well to the current literature discusssion. After a long time of weighing and considering, we have replaced it with “ecological livable city”.

This paper too many narrowly-focused case descriptions, and reads like better fit for a Chinese journal. For an international journal, the paper should have clearer theoretical and international contributions.

Reply: Many thanks for your constructive suggestion, we have supplemented theoretical and international marginal contributions in the Introduction from four aspects.

Line 27: “Guangdong-Hong Kong-Macao Greater Bay Area (GBA)” - start with international discussion rather than local cases, or international readers will lose interests immediately. There are abundant international discussions on livability and ecosystem services.

Reply: Many thanks for your constructive suggestion. We have rewritten the related contents, starting with an international discussion. The revisions are listed below.

The economy in the bay area is generally highly developed. Cities located in the bay area have experienced enormous economic growth and rapid urban transformation over the past few decades. The livability of bay area has attracted the attention of some scholars [7]. Guangdong-Hong Kong-Macao Greater Bay Area (GBA), New York Bay Area, San Francisco Bay Area, and Tokyo Bay Area are known as the four Bay Areas in the world.

Line 110: There is no literature within about 10 years, which highlight the need to modify the papers keyword choice.

 Reply: Many thanks for your constructive suggestion. We have added some literature within 10 years in the paper. Such as:

Furlan, R.; Petruccioli, A.; Major, M.D.; Zaina, S.; Zaina, S.; Al Saeed, M.; Saleh, D. The Urban Regeneration of West-Bay, Business District of Doha (State of Qatar): A Transit-Oriented Development Enhancing Livability. J. Urban Manag. 2019, 8(1), 126-144.

Wang, Y.; Zhang, H.; Wu, K. Spatial Differentiation and Influencing Factors of Housing Rents in the Guangdong-Hong Kong-Macao Greater Bay Area. Geogr. Res. 2020, 39(9), 2081-2094.

Cody, J.; Siravo, F. Historic Cities – Issues in Urban Conservation. Getty Conservation Institute: Los Angeles, USA, 2019.

Camerin, F. From “Ribera Plan” to “Diagonal Mar”, Passing through 1992 “Vila Olimpica”. How Urban Renewal Took Place as Urban Regeneration in Poblenou Distict (Barcelona). Land Use Policy 2019, 89, 10422.

Gale, D.E. Capital City: Gentrification and the Real Estate State. J. Plan Educ Res. 2019, 41(3), 359-361.

Sklair, L. The Icon Project: Architecture, Cities and Capitalist Globalization. Oxford University Press: New York, USA, 2017.

Fabris, L.M.F.; Camerin, F.; Semprebon, G.; Balzarotti, R.M. New Healthy Settlements Responding to Pandemic Outbreaks: Approaches from (and for) the Global City. The Plan J. 2020, 5(2), 385-406.

Line 200: “Therefore, this paper uses the principal component analysis method to evaluate the ecological livability of cities in the GBA”—the description is too rough. Please highlight the specifications of PCA.

 Reply: Many thanks for your constructive suggestion. We have added the specification of PCA. The corresponding descriptions are presented as follows:

To highlight the specification of PCA, we introduce some formal notation and nomenclature. X is a matrix of data with I rows and J columns. xj is the individual variable of X in the form of vector with I dimensions. t=w1x1+…+wJxJ=Xw.  indicates the squared Frobenius norm. The problem is

 (1)

To calculate how representative t is in terms of replacing X, regressing all variables of X on t is performed using the ordinary regression equation

              (2)

where p is the vector of coefficient and E is the matrix of residuals. The quality of t is judged by calculating .

From Equation (2), both t and p can be established by solving . The vector t is usually referred to as the score vector and the vector p is called the loading vector.

Equation 1: Equations should be illustrated in the method section, not in the result section.

 Reply: Many thanks for your constructive suggestion. We have deleted the contents related to Equation (1) in the result section, and add the following contents to the method section:

When evaluating the influencing factors of urban ecological livability, the following general dynamic spatial panel model [28] is established in this paper, and then the exploratory estimation of the model is carried out:

(3)

The explained variable is the ecological livability level of city i in year t, which is expressed by the comprehensive score of ecological livability;  is the first-order lag of the explained variable; is the spatial lag term of the explained variable;  is the influencing factor vector of ecological livability; is the spatial lag of the explained variable; is the individual effect of city i, and  is the time effect, is the line i of the error term space weight matrix. The spatial weight matrix is a symmetric matrix W which is composed of 0-1 elements based on the geographical adjacent relationship.

Line 366: “In this paper, three aspects of work are done.”—do not reiterate your method again. Be straightforward into the findings.  

Reply: Many thanks for your constructive suggestion. We have deleted “In this paper, three aspects of work are done.” The detailed deleted contents are as follows:

In this paper, three aspects of work are done. First, an evaluation index system of urban ecological livability is established for the GBA urban agglomeration, which takes into account the statistical differences caused by the institutional differences among the three regions. Second, the ecological livability of GBA urban agglomeration is evaluated comprehensively with the established evaluation index system of ecological livability by principal component analysis. Third, a spatial panel model is used to analyze the influencing factors of urban ecological livability, and the influencing factors are classified into three categories.

Reviewer 2 Report

The title of the article is Ecological Livability Assessment of Urban Agglomerations in Guangdong-Hong Kong-Macao Greater Bay Area. This paper has studied the ecological livability of GBA, and the factors that affect 11 the ecological livability of GBA. Issues related to ecological livability of the urban agglomeration is an interesting and very topical research topic.

The strengths of the manuscript are:

• selection of the topic, I believe that it fits into the research niche,

• correctly selected statistical methods,

• showing the research area on the map. 

A weakness of the manuscript is:

• no Discussion section,

• unequaled references.  

Please correct the following:

  1. In the introduction, please emphasize what is the innovation of the research.
  2. There is no data source under the tables and figures, even if it is an own study, it should be taken into account. Please add the source in all tables and graphs.
  3. Please correct the reference list like the first item: Liu, C .; Wang, T .; Guo, Q. Factors Aggregating Ability and the Regional Differences among China's Urban Agglomerations. 432 Sustainability 2018, 10, 4179. Items 6, 7, 8 and others are cited differently than required by the journal. Please correct.
  4. The "Discussion" section is missing. The article should be redrafted to separate such a section. The role of the discussion section is to discuss the findings (results) of your work, perhaps in comparison with previous results, available information, market conditions, etc.

Discussion: The limitations of the study are missing.

Discussion / conclusions: What are the policy implications of your findings? Please elaborate?

Discussion / conclusions: What are your suggestions for future research, based on the findings and limitations of your work?  

Author Response

Responses to Reviewer # 2

Comments and Suggestions for Authors

The title of the article is Ecological Livability Assessment of Urban Agglomerations in Guangdong-Hong Kong-Macao Greater Bay Area. This paper has studied the ecological livability of GBA, and the factors that affect 11 the ecological livability of GBA. Issues related to ecological livability of the urban agglomeration is an interesting and very topical research topic.

The strengths of the manuscript are:

  • selection of the topic, I believe that it fits into the research niche,
  • correctly selected statistical methods,
  • showing the research area on the map.

A weakness of the manuscript is:

  • no Discussion section,
  • unequaled references.  

Reply: Thank you for your objective and impartial comments. We have strengthed the weakness of the manuscript. In particular, we have added Discussion section and improved references. Please check it in the revision, and we appreciate all your work.

Please correct the following:

In the introduction, please emphasize what is the innovation of the research.

Reply: Many thanks for your constructive suggestion. We have emphasized the innovation of the research in the introduction. The corresponding paragraph is below.

The marginal contributions of this paper are summarized as follows. Firstly, we propose the “citizen-ecology-city” three-dimensional theoretical framework for urban ecological livability and establish the evaluation index system of urban ecological livability. The theoretical framework and the evaluation index system apply to other urban agglomerations in the world. Although there have been index systems in existing literature, the existing index systems are either not comprehensive or not practical for weak operability. It is necessary to develop a new practical index system to evaluate urban ecological livability scientifically and comprehensively, especially for urban agglomerations in bay areas like GBA, where two systems are implemented in a region, resulting in some statistical differences in indicators. According to the index system established in this paper, we can compare the ecological livability level of any two cities that belongs to socialist and capitalist countries respectively. Secondly, the ecological livability of urban agglomeration in GBA was evaluated quantitatively. Due to the difficulties in data collection and data processing process of unified indicator caliber, no relevant literature on the ecological livability of urban agglomerations in GBA has been found. Thirdly, through a spatial econometric model, we found that the ecological livability of GBA urban agglomeration has a significant spatial correlation. It enlightens us that the urban agglomeration of GBA is a closely linked whole, which means that we should not only strengthen economic and trade cooperation and people-to-people exchanges but also cooperate in ecological environment protection and green sustainable development closely. Fourth, through empirical analysis, we divide the factors of urban ecological livability into three categories: key factors, important factors, and general factors, which guide the government to take targeted measures to improve the ecological livability level of GBA urban agglomeration.

There is no data source under the tables and figures, even if it is an own study, it should be taken into account. Please add the source in all tables and graphs.

Reply: Many thanks for your constructive suggestion. We have added the data sources under the tables and figures.

Please correct the reference list like the first item: Liu, C .; Wang, T .; Guo, Q. Factors Aggregating Ability and the Regional Differences among China's Urban Agglomerations. 432 Sustainability 2018, 10, 4179. Items 6, 7, 8 and others are cited differently than required by the journal. Please correct.

Reply: Many thanks for your constructive suggestion. We have corrected the references.

The "Discussion" section is missing. The article should be redrafted to separate such a section. The role of the discussion section is to discuss the findings (results) of your work, perhaps in comparison with previous results, available information, market conditions, etc.

Discussion: The limitations of the study are missing.

Discussion / conclusions: What are the policy implications of your findings? Please elaborate?

Discussion / conclusions: What are your suggestions for future research, based on the findings and limitations of your work?  

Reply: Many thanks for your constructive suggestion. We have supplemented the “Discussion” section in the revision.

5.3. Discussion

Though the evaluation index system established in this paper has good application to the GBA through empirical study, it has not been used to analyze the ecological livability of other bay areas for the lack of data, such as the Tokyo Bay Area in Japan. In future research, a domestic or international comparative study of urban ecological livability can be conducted based on the evaluation index system of urban ecological livability. For the domestic comparison, we can compare GBA urban agglomeration with Yangtze River Delta urban agglomeration, Beijing-Tianjin-Hebei Urban Agglomeration, compare New York Bay Area urban agglomeration with San Francisco Bay Area. For the international comparison, we can compare GBA with Tokyo Bay Area, New York Bay Area, for example. More empirical researches will test the adequacy of the evaluation index system in this paper and verify the theoretical framework of urban ecological livability proposed.

Reviewer 3 Report

1.What is the main question addressed by the research?

The effective ecological livability of GBA, and the factors that affect the ecological livability of GBA

2. Do you consider the topic original or relevant in the field, and if so, why?

It is original and extremetly important today because human beings need to live in healthier, greener and more inclusive cities.

3. What does it add to the subject area compared with other published material?

This paper adds relevant findings in the field of liveable cities that may lead to comparison between the case study analyzed in this paper and other international case studies.

4. What specific improvements could the authors consider regarding the methodology? 

I found the need for a more comperhensive literature review at international level.

5. Are the conclusions consistent with the evidence and arguments presented and do they address the main question posed?

The conclusions are quite good.

6. Are the references appropriate?

The literature review is insufficient as the readership must understand the importance of the liveable cities by understanding the causes that hinder cities to be liveable. Add a paragraph about the causes and/or factors that make cities not well liveable in the international context.

  • the poor-designed redevelopment of historic neighborhoods, take as reference "2019. Historic Cities: Issues in Urban Conservation, The Getty Conservation Institute, Los Angeles"
  • urban planning has been used as a “weapon” for the capitalist interests (see 2019.  From “Ribera Plan” to “Diagonal Mar”, passing through 1992 “Vila Olímpica”. How urban renewal took place as urban regeneration in Poblenou district (Barcelona). Land Use Policy, 89, 10422). Mention the fact that behind the creation of a better urban environment (resilient and more sustainable), among the results of urban planning as generator and control mechanism there are social and economic polarization, rising rents and housing costs, people displacement (2019. Capital City. Gentrification and the real estate state. London-New York: Verso), high-standard new developments (2017. The icon project: architecture, cities and capitalist globalization. New York: Oxford University Press), and privatization.
  • unhealthy solutions for urban regeneration that are seemengly trying to overcome after the pandemic outbreak (take as reference "2020. New Healthy Settlements Responding to Pandemic Outbreaks. The Plan Journal, 5-2: 385-406". See also the report by OECD https://www.oecd.org/coronavirus/policy-responses/cities-policy-responses-fd1053ff/ 

7. Please include any additional comments on the tables and
figures.

Table 1 is not very understandable, put some space between the definitions.

To conclude, I need to revise the paper because of the lack in the literature review.

Author Response

Responses to Reviewer # 3

Comments and Suggestions for Authors

1.What is the main question addressed by the research?

The effective ecological livability of GBA, and the factors that affect the ecological livability of GBA

Reply: Thank you for your attentive reading.

  1. Do you consider the topic original or relevant in the field, and if so, why?

It is original and extremetly important today because human beings need to live in healthier, greener and more inclusive cities.

Reply: Thank you for your affirmation and positive comments.

  1. What does it add to the subject area compared with other published material?

This paper adds relevant findings in the field of liveable cities that may lead to comparison between the case study analyzed in this paper and other international case studies.

Reply: Many thanks for your affirmation and positive comments. We have emphasized this point in the description of marginal contributions in the introduction and discussion section.

  1. What specific improvements could the authors consider regarding the methodology?

I found the need for a more comperhensive literature review at international level.

Reply: Many thanks for your constructive suggestion. We have added some literature review in the paper. The supplemented literature review is listed below partly.

Livable cities are of great significance today because human beings need to live in healthier, greener, and more inclusive cities. However, some causes hinder cities to be livable. The poor-designed redevelopment of historic neighborhoods makes historic cities not well livable [12]. Urban planning has been used as a “weapon” for capitalist interests [13]. A desirable urban environment should be resilient and more sustainable. Behind the creation of a better urban environment, among the results of urban planning as a generator and control mechanism, there are social and economic polarization, rising rents and housing costs, people displacement [14], high-standard new developments [15], and privatization. Unhealthy solutions for urban regeneration are seemingly trying to be overcome after the pandemic outbreak [16,17], to make cities more livable. People all over the world hope to live in a livable urban and enjoy a high quality of life while factors that are not conducive to livability can be seen everywhere in the international context. We should know about the factors that make cities not livable and the factors that make cities livable.

  1. Are the conclusions consistent with the evidence and arguments presented and do they address the main question posed?

The conclusions are quite good.

Reply: We appreciate your rigorous work. And we have improved the conclusions further. The improved conclusions contains three parts: findings, policy implications, and Discussion. The corresponding contents are presented here.

5.1. Findings

Based on the above research work, the main findings are as follows. â‘  In GBA, the ecological livability of Hong Kong and Macao is much higher than that of the nine cities in the PRD, and first-tier cities Shenzhen and Guangzhou are in the leading position of the nine cities in the PRD. â‘¡ In the three dimensions that affect the ecological livability level of the GBA cities, from the perspective of citizens, people pay more attention to personal development opportunities and happiness; from the perspective of ecology, the living environment and ecosystem have a significant impact on the ecological livability; from the urban point of view, the comprehensive functions and development level of the city have a significant impact on the level of ecological livability. â‘¢ Although Hong Kong and Macao are crowded and people's living space is narrow, they are ecologically livable cities in terms of comprehensive perspective of civil development opportunities, ecological environment and ecosystem, urban comprehensive functions and economic and social development level.

5.2. Policy implications

According to the research findings above, this paper puts forward the following policy recommendations. â‘  According to the key factors affecting the ecological livable level of the city, efforts should be made to improve the ecological livable level of Jiangmen, Zhaoqing and other cities from three aspects of citizens, ecology and city, to improve the overall ecological livable level of GBA urban agglomeration. The results of ecological livability evaluation of GBA Urban Agglomeration show that the ecological livability levels of Jiangmen, Zhaoqing are at the bottom of GBA urban agglomeration, far below the average level, which seriously restricts the overall improvement of ecological livability level of GBA. Therefore, targeted measures should be taken to improve their ecological livability. â‘¡ For the nine cities in the PRD, economic development comes the first. Economic development can bring more livability [41]. Taking economic construction as the center, urban construction should learn from the advanced experience of Hong Kong and Macao. In terms of population density, Macao and Hong Kong are the most crowded cities in GBA, but this does not make them unlivable. The reason is that Macao and Hong Kong are the top two cities in GBA in terms of per capita GDP. Even from a global perspective, Macao and Hong Kong ranked fourth and 22nd in the global economy in 2018. As for per capita PPP GDP, Macao and Hong Kong ranked second and 11th in the global economy respectively in 2018 (World Bank). It can be seen that the high level of ecological livability is inseparable from the developed economy. For urban construction, taking infrastructure as an example, Macao and Hong Kong have a narrow geographical area, and the width of urban roads is generally narrower than that of nine cities in the PRD, but the frequency of traffic jams is not high, which shows that the operation efficiency of traffic infrastructure in Hong Kong and Macao is very high. Regarding drainage systems and sewage treatment systems, GBA is located in the south of China, where there is a lot of rain, but Hong Kong and Macao rarely suffer from flood disaster, and there is a relatively faultless sewage treatment and recycling system. In these aspects, nine cities in the PRD should learn from their successful experiences. â‘¢ While deepening the economic cooperation between Guangdong, Hong Kong, and Macao, we should strengthen the cooperation in ecological protection and environmental governance, and build a comfortable and livable world-class city group with green development, good ecology jointly. This paper finds that there is a spatial correlation between the level of ecological livability and its key influencing factors. Therefore, strengthening the cooperation of ecological environment protection among cities can promote each other and form a joint force, to realize the sustainable development of GBA better and faster.

5.3. Discussion

Though the evaluation index system established in this paper has good application to the GBA through empirical study, it has not been used to analyze the ecological livability of other bay areas for the lack of data. In future research, a domestic or international comparative study of urban ecological livability can be conducted based on the evaluation index system of urban ecological livability. For the domestic comparison, we can compare GBA urban agglomeration with Yangtze River Delta urban agglomeration, Beijing-Tianjin-Hebei Urban Agglomeration. For the international comparison, we can compare GBA with Tokyo Bay Area, New York Bay Area, and San Francisco Bay Area. More empirical researches will test the adequacy of the evaluation index system proposed in this paper and verify the theoretical framework of urban ecological livability proposed.

  1. Are the references appropriate?

The literature review is insufficient as the readership must understand the importance of the liveable cities by understanding the causes that hinder cities to be liveable. Add a paragraph about the causes and/or factors that make cities not well liveable in the international context.

the poor-designed redevelopment of historic neighborhoods, take as reference "2019. Historic Cities: Issues in Urban Conservation, The Getty Conservation Institute, Los Angeles"

urban planning has been used as a “weapon” for the capitalist interests (see 2019.  From “Ribera Plan” to “Diagonal Mar”, passing through 1992 “Vila Olímpica”. How urban renewal took place as urban regeneration in Poblenou district (Barcelona). Land Use Policy, 89, 10422). Mention the fact that behind the creation of a better urban environment (resilient and more sustainable), among the results of urban planning as generator and control mechanism there are social and economic polarization, rising rents and housing costs, people displacement (2019. Capital City. Gentrification and the real estate state. London-New York: Verso), high-standard new developments (2017. The icon project: architecture, cities and capitalist globalization. New York: Oxford University Press), and privatization.

unhealthy solutions for urban regeneration that are seemengly trying to overcome after the pandemic outbreak (take as reference "2020. New Healthy Settlements Responding to Pandemic Outbreaks. The Plan Journal, 5-2: 385-406". See also the report by OECD https://www.oecd.org/coronavirus/policy-responses/cities-policy-responses-fd1053ff/

Reply: Many thanks for your constructive suggestion. We have added a paragraph about the causes and/or factors that make cities not well livable in the international context. The paragraph added in the article is as follows.

Livable cities are of great significance today because human beings need to live in healthier, greener, and more inclusive cities. However, some causes hinder cities to be livable. The poor-designed redevelopment of historic neighborhoods makes historic cities not well livable [12]. Urban planning has been used as a “weapon” for capitalist interests [13]. A desirable urban environment should be resilient and more sustainable. Behind the creation of a better urban environment, among the results of urban planning as a generator and control mechanism, there are social and economic polarization, rising rents and housing costs, people displacement [14], high-standard new developments [15], and privatization. Unhealthy solutions for urban regeneration are seemingly trying to be overcome after the pandemic outbreak [16,17], to make cities more livable. People all over the world hope to live in a livable urban and enjoy a high quality of life while factors that are not conducive to livability can be seen everywhere in the international context. We should know about the factors that make cities not livable and the factors that make cities livable.

  1. Please include any additional comments on the tables and
    figures.

Table 1 is not very understandable, put some space between the definitions.

Reply: Many thanks for your constructive suggestion. We have put some space between the definitions in Table 1.

Table 1. Summary of definition of livable city and its related concepts.

Concept

Source of definition

Defining points

livable city

the long-term plan for Greater Vancouver (2003)

To bring physical, psychological and social benefits, personal development opportunities and rich spiritual and cultural wealth to the public; important principles: fairness, dignity, accessibility, cheerfulness, participation and rights protection [22].

Urban Livability

P. Evans (2002)

Survival and ecological sustainability: meet the survival needs of all citizens on the premise of protecting the ecological environment.

Livable city

D. Hahlweg (1997)

Healthy life, convenient travel, safe and charming, shared by all.

Livable city

E. Salzano (1997)

Connecting history and future, respecting historical footprints and future generations, sustainable development, providing material and social welfare, public space is the center of community and social life, and a network extending from downtown to suburb.

Livable city movement

Timothy D. Berg (1999)

Reshape the urban environment, build roads and blocks suitable for pedestrians, realize the comprehensive functions of the city such as work, residence and retail, and enhance the diversity of the city [23].

Livable City

H. L. Lennard (1997)

Citizens feel the existence of each other and face-to-face communication. Citizens are involved in many activities and celebrations. They feel safe. Public space can be used as a learning place and each citizen can be a learning object. It has multiple functions such as economy, society and culture. Citizens respect each other, the urban environment has aesthetic feeling, the opinions of citizens are respected and can participate in the process of urban development [24].

Evaluation index system of urban ecological livability

Lu et al (2012)

It covers the connotation of urban sustainable development strategic objectives, comprehensively reflects the stability and health of urban ecosystem, as well as all aspects of economic development, social development and ecological environment quality, objectively and truly reflects the urban ecological livability, and conforms to the concept, objectives and evaluation principles of urban sustainable development.

Livable City

Zhang (2016)

A livable city should be a city with a healthy environment, safety, pleasant nature, harmonious society, convenient life, and convenient travel [25].

Livable City

Liu et al (2019)

Livable City is a city with high quality of life and comfortable and pleasant living [26].

Source: The authors sorted out.

 To conclude, I need to revise the paper because of the lack in the literature review.

Reply: Many thanks for all your review work again.

Round 2

Reviewer 1 Report

Quality of life and ESV discussions are still very lacking. The current narrow focus won't receive many reading or citation. The language also needs significant improvement. I suggest using a language service. 

Author Response

Quality of life and ESV discussions are still very lacking. The current narrow focus won't receive many reading or citation. The language also needs significant improvement. I suggest using a language service. 

Reply: Many thanks for your suggestion. We have added the corresponding contents in the text and conducted a language service. Thanks again for your meticulous review and valuable suggestions to improve our manuscript. We hope that the revised revision has addressed all the issues. We are looking forward to your positive response.

Reviewer 3 Report

the paper is ready to be published.

Author Response

Reply:  Thanks again for your meticulous review and valuable suggestions to improve our manuscript.  God bless you!